# Joint Optimization on Trajectory, Cache Placement, and Transmission Power for Minimum Mission Time in UAV-Aided Wireless Networks

Tingting Lan, Danyang Qin * and Guanyu Sun

Department of Electronic and Communication Engineering, Heilongjiang University, Harbin 150080, China; 2191296@s.hlju.edu.cn (T.L.); 2191285@s.hlju.edu.cn (G.S.)
* Correspondence: qindanyang@hlju.edu.cn

**Abstract:** In recent years, due to the strong mobility, easy deployment, and low cost of unmanned aerial vehicles (UAV), great interest has arisen in utilizing UAVs to assist in wireless communication, especially for on-demand deployment in emergency situations and temporary events. However, UAVs can only provide users with data transmission services through wireless backhaul links established with a ground base station, and the limited capacity of the wireless backhaul link would limit the transmission speed of UAVs. Therefore, this paper designed a UAV-assisted wireless communication system that used cache technology and realized the transmission of multi-user data by using the mobility of UAVs and wireless cache technology. Considering the limited storage space and energy of UAVs, the joint optimization problem of the UAV's trajectory, cache placement, and transmission power was established to minimize the mission time of the UAV. Since this problem was a non-convex problem, it was decomposed into three sub-problems: trajectory optimization, cache placement optimization, and power allocation optimization. An iterative algorithm based on the successive convex approximation and alternate optimization techniques was proposed to solve these three optimization problems. Finally, in the power allocation optimization, the proposed algorithm was improved by changing the optimization objective function. Numerical results showed that the algorithm had good performance and could effectively reduce the task completion time of the UAV.

**Keywords:** UAV; trajectory; cache placement; transmission power

## 1. Introduction

Many countries are now using fifth-generation (5G) mobile communications in commercial deployments, and the industry, academia, and regulatory agencies have begun to research and develop next-generation mobile communication networks. As one of the access methods of the sixth-generation (6G) mobile communication network, UAVs are paramount to the goal of building an integrated air-space–ground–sea network to achieve emergency coverage, depth of coverage, and breadth of coverage [1–3]. Networked robotics and autonomous systems are typical applications of 6G, and UAV delivery systems are an example. Using drone technology in 6G would also help achieve cellular-free communications [4]. Therefore, UAVs will become important elements in 6G wireless communication, and, therefore, the research on UAV communication systems and related technologies has strategic significance [5].

At present, due to the high mobility, rapid deployment, flexible configuration, and the line-of-sight (LOS) link between UAVs and communication nodes in most cases, UAVs will play an important role in realizing high-speed wireless communications in communication systems [6]. In addition, UAVs can not only serve users on the ground as an air base station (BS), they also can reduce the data flow to the ground BSs in extremely crowded areas, enhance network reliability, and improve the quality of service [7,8]. In terms of mobile relay communication, there is almost no direct communication between the BSs on the

ground and the users. UAVs are, therefore, used to transmit information between the BSs and remote users [9]. Compared with a traditional ground-fixed relay, UAVs have more significant advantages as mobile relays [10]. However, for UAV-assisted wireless communications, in addition to consuming circuit power and transmitting power, the UAVs also consume a large amount of propulsion power to maintain flying or hovering at a certain height. Due to the limited energy of UAVs, it is necessary to optimize the deployment of UAVs and reasonably allocate the energy of UAVs.

At the same time, with the rapid development of multimedia services, communication between users has changed from traditional connection-centric communication to content-centric communication, that is the communication between users depends on large data files, such as videos and images, which bring great challenges to the future 5G/B5G networks. As a feasible technology to reduce network traffic load and improve network capacity, caching technology has been widely used in the field of cloud radio access networks (CRAN) [11]. In CRANs, edge nodes obtain popular content from the core network during off-peak hours and store it in their own hard disks, so that users can directly obtain the required content from the edge nodes, thus transferring the traffic in the backhaul link from peak to off-peak time periods, greatly reducing the traffic load on the backhaul link [12]. Since UAVs can only provide data transmission services to users through wireless backhaul links established with base stations on the ground, the limited capacity of the wireless backhaul link limits the transmission rate of UAVs and reduces the quality of service for the users [13]. UAV-assisted wireless communication using edge caching technology can effectively reduce the load of the wireless backhaul link, improve the performance of the UAV-assisted communication network, and provide users with a better quality of service. Therefore, cache-based UAV-assisted wireless communication is a technology with broad prospects for development, which can well meet the diverse and dynamic data requirements in the future 5G/B5G networks.

With the development of UAV technology and the popularization of civil UAV, UAVs have become an important means to collect geographic distribution data. For example, in the application of geographic information acquisition in a disaster, as the primary task of emergency support from a geographic information service, it is not only key to obtain the disaster distribution and disaster situation in a timely manner, but this is also the basis of emergency rescue, disaster assessment, post-disaster recovery, and reconstruction. In some areas, mountains and hills are widely distributed, the terrain uneven, and the climate complex and changeable. There are many difficulties in obtaining disaster information quickly after an emergency, so the emergency response capacity faces a major test. UAV systems as an important technical means of the rapid acquisition of disaster geographic information have been listed as an important part of the national aviation emergency rescue system. However, there are still many deficiencies in practice for UAV. The major technical problems in UAV disaster geographic information acquisition are the following: (1) the efficiency of UAV disaster information acquisition and on-site rapid processing is low under short-term airspace constraints; (2) the UAV's security, stability, and long-distance real-time transmission capability are insufficient in the context of complex conditions; (3) the capacity of the spatial analysis and dynamic simulation of disasters is poor. In terms of the geographic information industry, a UAV and geographic information system is far from the quality of aerial surveys. A UAV and geographic information system should be based on the applications of the GIS industry. The role of the UAV is not limited to aerial surveys, but it can also serve as an information collection platform. UAV aerial surveys emphasize the accuracy of the results, while GIS-oriented UAV applications emphasize reasonable cost, a flexible acquisition method, appropriate spatial accuracy, close to industry workflow, and good operability. This paper mainly focused on the deployment of emergency situations and temporary events in some complex and dangerous areas. To this end, this paper designed a cache-based UAV-assisted wireless communication network model, in which UAVs with a cache were used as mobile relays to provide data transmission services to users on the ground. The purpose of this research was to minimize the mission completion

time of the UAV under the constraints of meeting the maximum energy estimation of the UAV and the user data requirements. We summarized the main contributions of this paper as follows:

(1) Firstly, an mathematical model is developed for the UAV-assisted wireless communication using the cache technology. By optimizing the trajectory of the UAV, cache placement and the UAV transmit power, the data transmission service time of the UAV is minimized. In the optimization problem, this paper considers the maximum propulsion energy estimation, the user's minimum data requirements and other constraints and so on. In general, the formulation problem is non-convex, and it is difficult to get the optimal solution;

(2) Secondly, to solve the problem of non-convex optimization, an iterative (ITE) algorithm based on successive convex approximation and alternate optimization techniques are proposed. The formulated problem is non-convex with coupled variables. To facilitate the solution in a larger feasible region, this paper uses the slack variables to deal with the mathematical model of the optimization problem. Then it is divided into three sub-problems, which are cache placement optimization, UAV trajectory optimization, and UAV power optimization. Finally, SCA technology is used to solve three sub-optimal solutions. A sub-optimal solution of the non-convex problem was obtained by alternating solutions of cache placement optimization, UAV trajectory optimization, and UAV launch energy optimization;

(3) Finally, in the transmission power optimization process, an improved (IMP) algorithm is proposed by changing the optimization objective function. For the task completion time, there is no direct relationship with the transmission power of the UAV, but the more power the UAV allocates to the user, the higher the data transmission rate. Therefore, in the transmission power optimization process, the optimization objective function is changed to throughput maximization, where the cache placement and UAV trajectory are fixed. Simulation results show that the capacity-limited wireless backhaul link problem can be solved by optimizing cache placement, task completion time can be reduced by optimizing UAV trajectory, and system throughput can be maximized by optimizing transmit power. Through experimental comparison, the excellent performance of the improved algorithm is verified.

The rest of this paper is organized as follows. Section 2 introduces the system model. Section 3 introduces the formulation of minimizing task completion time problem. Section 4 proposes an effective iterative algorithm to solve the optimization problem and improves the algorithm. Section 5 presents the simulation results and some analysis. Finally, the conclusion is drawn in Section 6.

## 2. Related Work

At present, in the field of UAV-assisted wireless communication, many scholars have studied and discussed the typical problems of UAV communication systems, such as UAV trajectory, resource allocation, cache, and so on [14].

Trajectory optimization plays a very important role in UAV. By optimizing the flight trajectory of UAV, it has an important impact on improving the flight performance of UAV, and can also ensure the completion of flight tasks. In most realistic scenarios, various design states of UAV have been determined, and optimizing the flight trajectory of UAV is one of the few ways to improve the performance of UAV. Optimizing the trajectory of UAV can not only reduce the fuel consumption of the UAV, thereby further increasing the flight distance, but also shorten the flight time of UAV [15]. Zhang et al. [16] minimized the task completion time by optimizing the trajectory of the UAV. The minimum received signal-to-noise ratio was constrained during the entire mission flight to ensure the connection quality between the ground base station and the UAV link. Wu et al. [17] discussed some basic trade-offs between UAV communication and trajectory design. The result showed that the communication throughput, delay and propulsion energy consumption could be balanced by using different UAV trajectory designs, which provided a new idea in the traditional

ground communication. Zhang et al. [18] minimized the interrupt probability of the interrupted network by jointly optimizing the trajectory of the UAV and the transmitting power of the equipment. A low-complexity solution for the non-convexity problem was proposed. Under the requirement of meeting the data rate of all users, the total rate of edge users was maximized by optimizing the UAV trajectory and edge user scheduling strategy [19]. Aiming at the limited energy of UAV, Bian et al. [20] studied a UAV-assisted vehicle network, in which a UAV acted as an intermediary to communicate with a vehicle and the BS on the ground, respectively. In the case of multiple constraints, the trajectory and power allocation of UAV were jointly optimized.

With the development of wireless communication technology, the research on resource allocation has received more and more attention [21]. At present, the main consideration is how to allocate communication resources, such as transmission power and bandwidth to improve the performance of the communication system. In the wireless relay communication system, the power or energy of each node and the overall available bandwidth are limited. Therefore, under the limited resources, the research on power consumption and bandwidth is a very important topic, so optimizing the power and bandwidth allocation is an effective method to improve the energy efficiency of the system. Zhang et al. [22] studied the problem of secure communication and maximized the security rate of the system by optimizing the allocation of power allocation. Wu et al. [23] considered the power control to improve the energy efficiency performance of the system. Zhang et al. [24] minimized the UAV flight time while met the target rate requirements of each ground user by jointly optimizing the UAV's trajectory and the power and bandwidth allocation design methods. Wu et al. [25] studied an orthogonal frequency division multiple access (OFDMA) network supporting a UAV, in which the UAV was deployed as the BS. Serving a group of users on the ground and maximizing average throughput for all users by jointly optimizing the UAV's trajectory and the OFDMA resource allocation.

The UAV can only provide data transmission service to users through the wireless backhaul link established with the BS on the ground. The wireless backhaul link with a limited capacity will limit the transmission speed of the UAV, so it has a great challenge in aided wireless communication [26]. In order to reduce cellular data traffic, [27] researched a device-to-device-based UAV-assisted wireless network, in which the caching technology was applied to the UAV. Based on the two modes of the UAV static and dynamic, an optimization problem was designed to maximize the cache hit probability. In order to improve the quality of user experience, [28] studied a cache-based UAV-assisted wireless communication network. The author formulated a joint optimization problem about UAV cache placement and deployment to maximize the quality of user experience. In [29], Wu et al. maximized the network throughput by jointly optimizing cache and trajectory, and proposed a scheme based on deep supervised learning. For multimedia data, excessive data volume in communication is a major problem encountered by researchers. Ref. [30] proposed a UAV-assisted communication scheme using the cache technology, in which the location of the UAV and cache placement were jointly optimized to maximize system throughput.

Under the above background, based on the wireless cache technology, this paper designed a cache-based UAV-assisted wireless communication network model, in which the UAV with cache was used as a mobile relay to provide data transmission services for ground users. The purpose of this research was to minimize the mission completion time of the UAV under the constraints of meeting the maximum energy estimation of the UAV and the user data requirements.

## 3. System Model

This paper designs a cache-based UAV-assisted wireless communication scheme, in which the BS and the UAV cooperate to serve multiple ground users. The system model is shown in Figure 1. In this paper, we assume that there are $U$ ground users, using the set $UU = \{1, 2 \ldots, U\}$, the horizontal position of user $u$ is represented by $Z_u = [x_u, y_u], u \in UU$,

where the position of each user is known in this paper. In this paper, the UAV with cache technology performs the flight mission according to the designed trajectory at the fixed altitude $H$. Define the task completion time of the UAV as $T$, and the horizontal trajectory coordinate at time $t$ is denoted as $L_u(t) \in R^2, 0 \leq t \leq T$. In addition, $v(t) \triangleq \dot{L}_u(t)$ is defined as the speed of the UAV at time $t$, where $\dot{L}_u(t)$ represents the derivative of $L_u(t)$ with respect to time $t$, and $V_{max}$ is the maximum speed of the UAV, so $\|\dot{L}_u(t)\| \leq V_{max}, \forall t \in [0, T]$. However, the continuous variable $t$ means that there will have infinite speed constraints, which is not conducive to the subsequent solution. Therefore, the discrete trajectory approximation technique can generate a finite number of variables and constraints. In this paper, the UAV task completion time $T$ is discretized into $N$ equal interval time slots which are small enough, namely $T = N\delta_t$, where $\delta_t$ represents the length of time slot. Based on this discretization, UAV trajectory $L_u(t)$ can be represented by sequence $\{L_u(n), 1 \leq n \leq N\}$. $L[1] = L_I$ is the initial position of the UAV, and $L[N+1] = L_D$ is the destination of the UAV when the mission is completed. The trajectory of UAV is limited by the maximum speed and can be expressed as

$$\|v[n]\| \triangleq \frac{\|L_u[n+1] - L_u[n]\|}{\delta_t} \leq V_{max}, 1 \leq n \leq N-1 \tag{1}$$

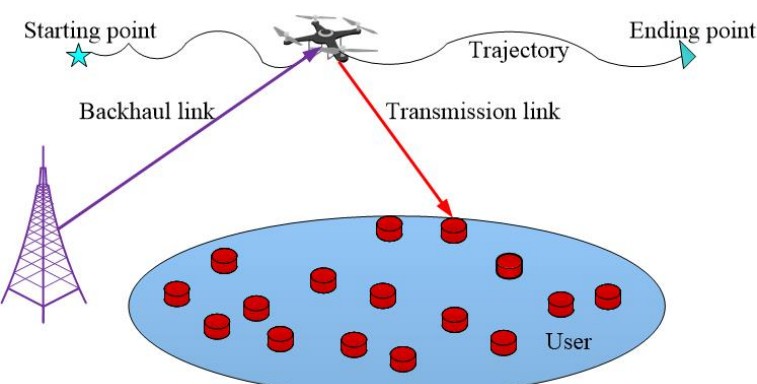

**Figure 1.** System model.

### 3.1. Channel Model

Generally, $h_u[n]$ denotes the channel coefficient between the UAV and the user $u$ in the $n$th time slot, i.e., $h_u[n] = \tilde{h}_u[n] \cdot \sqrt{\beta_u[n]}$, where $\beta_u[n]$ denotes large-scale fading effects such as the path loss and the shadow, and $\tilde{h}_u[n]$ denotes the complex valued random variable. In time slot $n$, the distance from the UAV to the user $u$ is $D_u[n] = \sqrt{H^2 + \|L_u[n] - Z_u\|^2}, \forall u \in UU, n \in N$. Assuming that the wireless channel between the user $u$ and the UAV is controlled by the line of sight (LoS), and is based on the free space path loss model, the power gain of the LoS channel from the UAV to the user $u$ in the $n$th slot can be expressed as [31]

$$\beta_u[n] = \beta_0 D_u^{-2}[n] = \frac{\beta_0}{(H^2 + \|L_u[n] - Z_u\|^2)^{\alpha/2}} \tag{2}$$

where $\alpha$ represents the path loss index, $\beta_0$ represents the path loss at the reference distance.

Generally, the probability of the LoS depends on the statistical model of the propagation environment, the building density, and height. Therefore, in the $n$th time slot, the LoS link probability between the UAV and the user $u$ is [32]

$$P_{LoS}^u[n] = \frac{1}{1 + b_1 exp(-b_2[m_u[n] - b_1])} \tag{3}$$

where the two parameter values of $b_1$ and $b_2$ are determined by the external environment. $m_u[n]$ is the elevation angle between the UAV and the user $u$. $P_u[n]$ is expressed as the

transmission power of UAV, and $0 \leqslant P_u[n] \leqslant P_{max}, \forall n$. Where $P_{max}$ is the maximum transmission power. In time slot $n$, when the UAV transmits data to the user $u$, the signal-to-noise ratio (SNR) received by the user $u$ is defined as

$$\gamma_u[n] = \frac{P_u[n]\beta_u[n]}{\sigma_u^2} \tag{4}$$

where $\sigma_u^2$ is the noise power spectral density at the receiver.

### 3.2. Cache Placement Model

In this paper, a UAV based on caching technology is used to carry out the data transmission service. During the non-peak traffic hours, the UAV can cache some popular contents. There are $F$ contents stored in the BS, and the content sequence is expressed as $\{1, 2 \ldots, F\}$ and each content is of equal size. The content popularity sequence is expressed as $f \in \{f_1, f_2, \ldots f_F\}$, and the fth content popularity is defined as the probability that the $f$th content is requested by the user. The following expression can be obtained from the Zipf distribution [33].

$$P_f = \frac{f^{-\varsigma}}{\sum_{j=1}^{F} j^{-\varsigma}} \tag{5}$$

where $\varsigma$ is the Zipf distribution parameter.

In order to achieve a higher hit rate under limited storage space, this paper adopts the most commonly used caching strategy, that is, the most popular content caching strategy. Due to the limitation of capacity, the UAV can only cache some popular contents. The definition $C_r$ represents the number of contents stored by the UAV, and $C_r \leqslant F$. In time slot $n$, the user $u$ can request popular content independently, and the probability of requesting content $f$ is denoted as $P_u^f[n] = \frac{f^{-\varsigma}}{\sum_{i=1}^{F} i^{-\varsigma}}$, and $0 \leqslant P_u^f[n] \leqslant 1$. $I = \{I_u^f[n] \in \{0,1\}\}, u \in UU$ is the cache placement vector of each content. In the $n$ time slot, $I_u^f[n] = 1$ indicates that the user $u$ can obtain the required content $f$ from the UAV. In addition, the UAV can only serve one user in a period of time, and the cache constraint is expressed as follows.

$$I_u^f[n] \in \{0, 1\}, \forall u, f, n \tag{6}$$

$$\sum_{u=1}^{U} I_u^f[n] \leqslant 1, \forall f, n \tag{7}$$

$$\sum_{f=1}^{F} I_u^f[n] \leqslant C_r, \forall u, n \tag{8}$$

If the content requested by the user exists in the UAV, the UAV will directly transmit the content to the user without communicating with the BS, thereby reducing the transmission delay and improving the quality of service of the user; if the UAV does not cache the content requested by the user, the UAV will send a request to the BS, and the BS will first transmit the content to the UAV, and then the UAV will transmit the content to the user. In time slot $n$, the transmission rate of the $f$th content transmitted by the UAV to the user $u$ as

$$\begin{aligned}
R_u^f[n] &= P_u^f[n] I_u^f[n] log_2(1 + \gamma_u[n]) \\
&= P_u^f[n] I_u^f[n] log_2(1 + \frac{P_u[n]\beta_u[n]}{\sigma_u^2}) \\
&= P_u^f[n] I_u^f[n] log_2(1 + \frac{\phi_u[n]}{(H^2 + \|L_u[n] - Z_u\|^2)^{\alpha/2}}), \forall n
\end{aligned} \tag{9}$$

where $\phi_u[n] = P_u[n]\beta_0/\sigma_u^2$.

In this paper, we express the amount of data and data requirement constraint that user $u$ can receive in the $n$th slot as follows.

$$\sum_{n=1}^{N} \delta_t R_u^f[n] \geq S_u, \forall u \tag{10}$$

the left part of the constraint (10) defines the data that the UAV transmits to the user after receiving the user's request. Where $S_u$ represents the data requirements of each user within the completion time of the UAV task.

### 3.3. Energy Consumption Model

In many practical scenarios, the energy consumption during the flight of the UAV is far greater than communication energy consumption of the UAV [34], so this article only considers the flight energy consumption of the UAV. The UAV flight power is expressed as [20].

$$P(\|v[n]\|) = P_0(1 + \frac{3\|v[n]\|^2}{U_{tip}^2}) + P_1(\sqrt{1 + \frac{\|v[n]\|^4}{4v_0^4}} - \frac{\|v[n]\|^2}{2v_0^2})^{0.5} + \frac{1}{2}d_0 \rho s A \|v[n]\|^3, \forall n \tag{11}$$

where $P_0 = \frac{\delta}{8} \rho s A \Omega^3 R^3$ represents the power of the blade profile and $P_1 = (1+k)\frac{G_g^{1.5}}{\sqrt{2\rho A}}$ is the induced power in the hovering state. $\Omega$ represents the angular velocity of the blade and $G_g$ is the weight of the UAV. $\delta$ is the profile drag coefficient. $U_{tip}$ represents the blade tip velocity and $v_0$ is the average blade induced hovering speed. $k$ is a correction factor. $d_0$ is the drag ratio of the fuselage. $s$ is the rotor stiffness. $A$ is the rotor disk area and $\rho$ is the air density. Therefore, the total flight energy consumption of UAV is expressed as [35]

$$E = \sum_{n=1}^{N} \delta_t(P_0 + \frac{3P_0\|v[n]\|^2}{U_{tip}^2} + \frac{1}{2}d_0 \rho s A \|v[n]\|^3) + \sum_{n=1}^{N} \delta_t P_1(\sqrt{1 + \frac{\|v[n]\|^4}{4v_0^4}} - \frac{\|v[n]\|^2}{2v_0^2})^{0.5} \tag{12}$$

Defining $O_{max}$ as the energy carried by the UAV, and the constraint is expressed as

$$E \leq O_{max} \tag{13}$$

## 4. Problem Formulation for Time Minimization

In order to minimize the task completion time of the UAV, the following optimization problem is established by optimizing the UAV trajectory $L = \{L_u[n], \forall n\}$, cache placement $I = \{I_u^f[n], \forall n, u, f\}$ and transmission power $P = \{P_u[n], \forall n, u\}$.

$$(P1): \min_{\delta_t, L, I, P} T$$

$$s.t. \sum_{n=1}^{N} \delta_t R_u^f[n] \geq S_u, \forall u, f \tag{14}$$

$$I_u^f[n] \in \{0, 1\}, \forall u, f, n \tag{15}$$

$$\sum_{u=1}^{U} I_u^f[n] \leqslant 1, \forall f, n \tag{16}$$

$$\sum_{f=1}^{F} I_u^f[n] \leqslant C_r, \forall u, n \tag{17}$$

$$L[1] = L_I, L[N+1] = L_D \tag{18}$$

$$\frac{\|L_u[n+1] - L_u[n]\|}{\delta_t} \leq V_{max}, 1 \leq n \leq N - 1 \tag{19}$$

$$E \leq O_{max} \tag{20}$$

$$0 \leqslant P_u[n] \leqslant P_{max}, \forall n \tag{21}$$

Constraint (14) ensures that the UAV can transmit data to the user in any time slot $n$. Constraints (15)–(17) ensure that the content requested by the user exists in the local cache of the UAV. (18) and (19) are about the trajectory constraint of the UAV. In terms of the speed of the UAV, constraint (19) also restricts the maximum speed and (18) represents the starting point and ending point of UAV. Constraint (20) ensures that the UAV can complete data transmission service under limited energy. (21) restricts the transmission power of the UAV.

First, (15) is an integer constraint because there is a binary variable in the constraint (15). Second, the inequalities in (14) and (20) have mutually coupled variables, which is a non-convex constraint. Therefore, due to the above reasons, the optimization problem (P1) in this paper is a mixed-integer non-convex problem, which can not be effectively solved by the existing methods.

## 5. Problem Solution

In order to make the problem (P1) easier to deal with, the binary variable in (15) is relaxed into a continuous variable, and then the optimization problem (P1) can be represented by (P2).

$$(P2): \min_{\delta_t, L, I, P} T$$

$$s.t. \sum_{n=1}^{N} \delta_t R_u^f[n] \geq S_u, \forall u, f \tag{22}$$

$$0 \leqslant I_u^f[n] \leqslant 1, \forall u, f, n \tag{23}$$

$$\sum_{u=1}^{U} I_u^f[n] \leqslant 1, \forall f, n \tag{24}$$

$$\sum_{f=1}^{F} I_u^f[n] \leqslant C_r, \forall u, n \tag{25}$$

$$L[1] = L_I, L[N+1] = L_D \tag{26}$$

$$\frac{\|L_u[n+1] - L_u[n]\|}{\delta_t} \leq V_{max}, 1 \leq n \leq N-1 \tag{27}$$

$$E \leq O_{max} \tag{28}$$

$$0 \leqslant P_u[n] \leqslant P_{max}, \forall n \tag{29}$$

This relaxation generally means that the objective value of the problem (P2) is represented as the upper bound of the objective value of the problem (P1). Although the problem (P2) is relaxed, it is still a non-convex optimization problem due to the existence of non-convex constraints. Next, we propose an efficient iterative algorithm for the non-convex problem (P2) by using the successive convex approximation and the alternating optimization techniques. The core idea is to solve the three sub-problems of (P2) iteratively, that is, to optimize the cache placement by fixing the trajectory and transmission power of the UAV; The trajectory of the UAV is optimized by fixing the cache placement and the transmission power of the UAV; to optimize the transmission power of the UAV by fixing UAV trajectory and cache placement.

### 5.1. Cache Placement Optimization

After fixing the trajectory and transmission power of the UAV, the cache placement is optimized by solving the following problem (P3).

$$(P3): \min_{\delta_t, I} T$$

$$s.t. \sum_{n=1}^{N} \delta_t R_u^f[n] \geq S_u, \forall u, f \tag{30}$$

$$0 \leqslant I_u^f[n] \leqslant 1, \forall u, f, n \tag{31}$$

$$\sum_{u=1}^{U} I_u^f[n] \leqslant 1, \forall f, n \tag{32}$$

$$\sum_{f=1}^{F} I_u^f[n] \leqslant C_r, \forall u, n \tag{33}$$

Since the problem (P3) is a standard linear programming form, the cache layout can be optimized by solving the linear programming, and the existing optimization tools (such as CVX) can effectively solve the convex optimization problem.

### 5.2. UAV Trajectory Optimization

After giving cache placement and the transmission power of the UAV, the UAV's trajectory is optimized by solving the following problem (P4).

$$(P4): \min_{\delta_t, L} T$$

$$s.t. \sum_{n=1}^{N} \delta_t R_u^f[n] \geq S_u, \forall u, f \tag{34}$$

$$L[1] = L_I, L[N+1] = L_D \tag{35}$$

$$\frac{\|L_u[n+1] - L_u[n]\|}{\delta_t} \leq V_{max}, 1 \leq n \leq N-1 \tag{36}$$

$$E \leq O_{max} \tag{37}$$

Since constraint (34) and (37) are both non-convex, problem (P4) is still a non-convex optimization problem. The slack variables $Y \triangleq \{y_u[n]\}$ and $O \triangleq \{o_u[n] \geqslant 0\}$ are introduced, respectively. Where $o_u[n] = (\sqrt{1 + \frac{\|v[n]\|^4}{4v_0^4}} - \frac{\|v[n]\|^2}{2v_0^2})^{0.5}$, which is equivalent to $\frac{1}{o_u[n]^2} = o_u[n]^2 + \frac{\|v[n]\|^2}{2v_0^2}$. Based on the above, there are the following expressions.

$$(P5): \min_{\delta_t, L, Y, O} T$$

$$s.t. R_u^f[n] \geq y_u[n], \forall u, f, n \tag{38}$$

$$\sum_{n=1}^{N} \delta_t y_u[n] \geq S_u, \forall u, f \tag{39}$$

$$o_u[n]^2 + \frac{\|v[n]\|^2}{2v_0^2} \geq \frac{1}{o_u[n]^2} \tag{40}$$

$$o_u[n] \geq 0 \tag{41}$$

$$\sum_{n=1}^{N} \delta_t \left( P_0 + \frac{3P_0\|v[n]\|^2}{U_{tip}^2} + \frac{1}{2}d_0\rho sA\|v[n]\|^3 \right) + \sum_{n=1}^{N} \delta_t P_1 o_u[n] \leq O_{max} \tag{42}$$

$$L[1] = L_I, L[N+1] = L_D \tag{43}$$

$$\frac{\|L_u[n+1] - L_u[n]\|}{\delta_t} \leq V_{max}, 1 \leq n \leq N-1 \tag{44}$$

Because there are new non-convex constraints in (38)–(40), (P5) is still a non-convex problem. In order to solve this problem, the slack variables $J = \{j_u[n], \forall u, n\}$ are introduced to deal with $R_u^f[n]$. There are the following questions (P6).

$$(P6): \min_{\delta_t, L, Y, O, J} T$$

$$s.t. \|L_u[n] - Z_u\|^2 \geq j_u[n], \forall u, n \tag{45}$$

$$R_u^f[n] = P_u^f[n] I_u^f[n] log_2 (1 + \frac{\phi_u[n]}{(H^2 + j_u[n])^{\alpha/2}}) \geq y_u[n], \forall u, f, n \tag{46}$$

$$\sum_{n=1}^{N} \delta_t y_u[n] \geq S_u, \forall u, f \tag{47}$$

$$o_u[n]^2 + \frac{\|v[n]\|^2}{2v_0^2} \geq \frac{1}{o_u[n]^2} \tag{48}$$

$$o_u[n] \geq 0 \tag{49}$$

$$\sum_{n=1}^{N} \delta_t (P_0 + \frac{3P_0 \|v[n]\|^2}{U_{tip}^2} + \frac{1}{2} d_0 \rho s A \|v[n]\|^3) + \sum_{n=1}^{N} \delta_t P_1 o_u[n] \leq O_{max} \tag{50}$$

$$L[1] = L_I, L[N+1] = L_D \tag{51}$$

$$\frac{\|L_u[n+1] - L_u[n]\|}{\delta_t} \leq V_{max}, 1 \leq n \leq N-1 \tag{52}$$

By solving the problem (P6), we can get the optimal solution of the problem (P5). In (P6), $J$ is convex to $\|L_u[n] - Z_u\|^2$ in constraint (45). Since there are non-convex constraints in (45) and (47), then (P6) is still non-convex. At a given local point, using SCA technology, each iteration can convert the function that needs to be processed into a form that is easy to solve. Based on the given local points $\delta_t^r$ and $y_u^r[n]$, using the first-order Taylor (FoT) expansion of $(\delta_t + y_u[n])^2$ in constraint (47), the global lower bound can be obtained.

$$(\delta_t + y_u[n])^2 \geq -(\delta_t^r + y_u^r[n])^2 + 2(\delta_t^r + y_u^r[n]) \times (\delta_t + y_u[n]), \forall u \tag{53}$$

It can be seen from (54).

$$\begin{aligned} \delta_t y_u[n] &= \frac{(\delta_t + y_u[n])^2 - (\delta_t^2 + y_u^2[n])}{2} \\ &\geq (\delta_t^r + y_u^r[n])(\delta_t + y_u[n]) - \frac{\delta_t^2 + y_u^2[n]}{2} - \frac{(\delta_t^r + y_u^r[n])^2}{2} \\ &\triangleq g_u[n] \end{aligned} \tag{54}$$

where $g_u[n]$ is convex relative to $\delta_t$ and $y_u[n]$.

In constraint (48), $\|v[n]\|^2$ is convex for $v[n]$, and $o_u[n]^2$ is also a convex function of $o_u[n]$. Given local points $v^r[n]$ and $o_u^r[n]$, $o_u[n]^2 + \frac{\|v[n]\|^2}{2v_0^2}$ is applied to the FoT expansion to obtain the following inequality.

$$\begin{aligned} o_u[n]^2 + \frac{\|v[n]\|^2}{2v_0^2} &\geq o_u^r[n]^2 + 2o_u^r[n](o_u[n] - o_u^r[n]) + \|v^r[n]\|^2 + 2(v^r[n])^T(v[n] - v^r[n]) \\ &\triangleq w_u[n] \end{aligned} \tag{55}$$

The $w_u[n]$ is a linear function of $v[n]$ and $o_u[n]$. Similarly, for constraint (45), apply a first-order Taylor expansion to $\|L_u[n] - Z_u\|^2$ at a given point $L_u^r[n]$.

$$\|L_u[n] - Z_u\|^2 \geq \|L_u^r[n] - Z_u\|^2 + 2(L_u^r[n] - Z_u)^T \times (L_u[n] - L_u^r[n]) \tag{56}$$

$R_u^f[n]$ is convex with respect to $\|L_u[n] - Z_u\|^2$ in constraint (46). When the local point $L_u^r[n]$ is given, the lower bound in the $r$th iteration is obtained by expanding $R_u^f[n]$ through the FoT.

$$R_u^f[n] = P_u^f[n]I_u^f[n]log_2(1 + \frac{\phi_u[n]}{(H^2 + \|L_u^r[n] - Z_u\|^2)^{\alpha/2}})$$
$$\geq \hat{R}_u^f[n] \triangleq Q_u^r[n] - A_u^r[n](\|L_u[n] - Z_u\|^2 - \|L_u^r[n] - Z_u\|^2) \quad (57)$$

where

$$A_u^r[n] = \frac{P_u^f[n]I_u^f[n]\phi_u[n]log_2e}{(H^2 + \|L_u^r[n] - Z_u\|^2)(H^2 + \|L_u^r[n] - Z_u\|^2 + \phi_u[n])} \quad (58)$$

$$Q_u^r[n] = P_u^f[n]I_u^f[n]log_2(1 + \frac{\phi_u[n]}{H^2 + \|L_u^r[n] - Z_u\|^2}) \quad (59)$$

For any given local points $\delta_t$, $y_u^r[n]$, $v^r[n]$, $o_u^r[n]$, $L_u^r[n]$ and the lower bound of (54)–(57), the problem (P6) is represented by (P7).

$(P7): \min_{\delta_t,L,Y,O,J,V} T$

$s.t. \sum_{n=1}^{N} \delta_t g_u[n] \geq S_u, \forall u \quad (60)$

$w_u[n] \geq \frac{1}{o_u[n]^2}, \forall u, n \quad (61)$

$\|L_u^r[n] - Z_u\|^2 + 2(L_u^r[n] - Z_u)^T \times (L_u[n] - L_u^r[n]) \geq j_u[n], \forall u, n \quad (62)$

$\hat{R}_u^f[n] \geq y_u[n], \forall u, n \quad (63)$

$L[1] = L_I, L[N+1] = L_D \quad (64)$

$\frac{\|L_u[n+1] - L_u[n]\|}{\delta_t} \leq V_{max}, 1 \leq n \leq N - 1 \quad (65)$

$o_u[n] \geq 0 \quad (66)$

The analysis shows that (P7) is a convex optimization problem, which can be solved using CVX.

*5.3. UAV Transmission Power Optimization*

After giving the cache placement and the UAV's trajectory, optimize the UAV's transmission power by solving the following problems (P8).

$(P8): \min_{\delta_t,P} T$

$s.t. \sum_{n=1}^{N} \delta_t R_u^f[n] \geq S_u, \forall u \quad (67)$

$E \leq O_{max} \quad (68)$

$0 \leqslant P_u[n] \leqslant P_{max}, \forall u, n \quad (69)$

It can be seen from the above expression that the problem (P8) is convex for the transmission power $P \triangleq \{P_u[n]\}$. However, we convert the constraint (67) into another form, which is easier to handle. After introducing variables $\{\lambda_u[n], \forall u, n\}$ and defining $\chi_u[n] = \delta_t \lambda_u[n] = \frac{1}{2}(\delta_t + \lambda_u[n])^2 - \frac{1}{2}(\delta_t^2 + \lambda_u^2[n])$. At this time, the constraint (67) can be expressed as.

$$\sum_{n=1}^{N} \chi_u[n] \geq S_u \quad (70)$$

$$\lambda_u[n] \leq R_u^f[n], \forall u, n \tag{71}$$

Next, the lower bound of $\chi_u[n]$ is obtained by first-order Taylor expansion, namely $\tilde{\chi}_u[n] = -\frac{1}{2}(\delta_t^2 + \lambda_u^2[n]) + (\delta_t^r + \lambda_u^r[n])(\delta_t + \lambda_u[n]) - \frac{1}{2}(\delta_t^r + \lambda_u^r[n])^2$. Constraint (70) can be expressed as $\sum_{n=1}^N \tilde{\chi}_u[n] \geq S_u$. Introduce the slack variable$\{\xi_u[n], \forall u, n\}$, and define the lower bound of $\lambda_u[n]$ as $\xi_u[n]$, namely $\xi_u[n] = log_2(1 + 1/\xi_u[n]) - (\xi_u[n] - \xi_u^r[n])/(ln2\xi_u^r[n](\xi_u^r[n] + 1))$. Constraint (71) can be restated as

$$\lambda_u[n] \leq \xi_u[n], \forall u, n \tag{72}$$

$$\|P_u[n], \xi_u[n], (\frac{2}{(\beta_0/\sigma_u^2)/((H^2 + \|L_u[n] - Z_u\|^2)^{\alpha/2})})^{0.5}\| \leq P_u[n] + \xi_u[n], \forall u, n \tag{73}$$

For the non-convex term in constraint (68), let $\eta_u[n] = \|L_u[n+1] - L_u[n]\|$, then $\|v[n]\| \triangleq \frac{\eta_u[n]}{\delta_t}$, and at the same time introduce a slack variable $c_u[n] \geq 0$ to transform it into a more tractable form.

$$c_u[n] = (\sqrt{\delta_t^4 + \frac{\eta_u^4[n]}{4v_0^4}} - \frac{\eta_u^2[n]}{2v_0^2})^{0.5} \tag{74}$$

therefore, the constraint (67) can be transformed into the following form.

$$\sum_{n=1}^N (\delta_t P_0 + \frac{3P_0\eta_u^2[n]}{U_{tip}^2\delta_t} + \frac{1}{2\delta_t}d_0\rho s A\eta_u^3[n]) + \sum_{n=1}^N P_1 c_u[n] \leq O_{max}, \forall u,, n \tag{75}$$

$$c_u[n] \geq (\sqrt{\delta_t^4 + \frac{\eta_u^4[n]}{4v_0^4}} - \frac{\eta_u^2[n]}{2v_0^2})^{0.5}, \forall u,, n \tag{76}$$

$$c_u[n] \geq 0, \forall u,, n \tag{77}$$

The constraint (76) can satisfy the constraint (75) by continuously reducing the value of the slack variable, but (76) is still non-convex. In order to facilitate processing, the constraint (76) is transformed into the following form.

$$\frac{\delta_t^4}{c_u^2[n]} \leq c_u^2[n] + \frac{\eta_u^2[n]}{v_0^2} \tag{78}$$

Note that the inequality constraint (78) is still non-convex. The right-hand side of (78) is transformed by first-order Taylor expansion into:

$$\frac{\delta_t^4}{c_u^2[n]} \leq c_u^{2r}[n] + 2c_u^r[n](c_u[n] - c_u^r[n]) + \frac{\|\eta_u^r[n]\|}{v_0^2} + \frac{2\eta_u^r[n]}{v_0^2}(\eta_u[n] - \eta_u^r[n]) \tag{79}$$

where $c_u^r[n]$ and $\eta_u^r[n]$ are the values of $c_u[n]$ and $\eta_u[n]$ at the $r$th iteration. Finally, the transmission power optimization problem can be solved by the following convex problem.

$$(P9): \min_{\delta_t, P_u[n], \lambda_u[n], c_u[n], \xi_u[n]} T$$

$$s.t. \sum_{n=1}^{N} \tilde{\chi}_u[n] \geq S_u, \forall u \tag{80}$$

$$0 \leqslant P_u[n] \leqslant P_{max} \tag{81}$$

$$\lambda_u[n] \leq \xi_u[n], \forall u, n \tag{82}$$

$$c_u[n] \geq 0 \tag{83}$$

$$\frac{\delta_t^4}{c_u^2[n]} \leq c_u^{2r}[n] + 2c_u^r[n](c_u[n] - c_u^r[n]) + \frac{\|\eta_u^r[n]\|}{v_0^2} + \frac{2\eta_u^r[n]}{v_0^2}(\eta_u[n] - \eta_u^r[n]) \tag{84}$$

$$\sum_{n=1}^{N} (\delta_t P_0 + \frac{3P_0\eta_u^2[n]}{U_{tip}^2 \delta_t} + \frac{1}{2\delta_t} d_0 \rho s A \eta_u^3[n]) + \sum_{n=1}^{N} P_1 c_u[n] \leq O_{max}, \forall u, n \tag{85}$$

$$\|P_u[n], \xi_u[n], (\frac{2}{(\beta_0/\sigma_u^2)/((H^2 + \|L_u[n] - Z_u\|^2)^{\alpha/2})})^{0.5}\| \leq P_u[n] + \xi_u[n], \forall n \tag{86}$$

By solving the above convex problems, $\delta_t$, $P_u[n]$, $\lambda_u[n]$, $\xi_u[n]$ and $c_u[n]$ can be updated, and it can be effectively solved by CVX.

### 5.4. Joint Flight Trajectory, Cache Placement, and Transmission Power Optimization

Based on the discussion in the previous three sections, this section proposes the ITE algorithm to jointly optimize the trajectory, cache placement, and transmission power of the UAV. The specific iteration process is shown in Algorithm 1.

---

**Algorithm 1** ITE algorithm

---

**Initialize**: $L^0, P^0, I^0$, and let $\delta_t^0, r = 0$.
1: **Repeat**
2:　　　Fix $\{L, P\}$, obtain the optimal cache placement as $I_u^{f*}[n]$ by solving (P3);
3:　　　Fix $\{I, P\}$, obtain the optimal trajectory as $L_u^*[n]$ by solving (P4);
4:　　　Fix $\{I, L\}$, obtain the optimal power allocation as $P_u^*[n]$ by solving
　　　　(P8);
5: **Update** : $r = r + 1$
6: **Until**:　Converge to a prescribed accuracy.
7: **Output**:　the Cache placement $I$, Transmission power $P$, UAV trajectory $L$ and
　　　　　　completion time $T$.

---

In the above ITE algorithm, after fixing the cache placement and the trajectory of the UAV, the optimization of the transmission power is mainly to minimize the objective function by optimizing the speed of the UAV. Due to the existence of constraints (27) and (28), when the UAV's trajectory is optimized, the UAV's speed will reach the maximum value. So if there are no energy and speed constraints in the ITE algorithm, the performance will be better. Since the problem (P9) has many optimization variables, it not only increases the computational complexity, but also easily falls into the local optimal value. In fact, the relationship between the optimization of the objective function and the transmission power of the UAV is not very obvious. However, when the UAV is transmitting data, the more power allocated to the user, the higher the data transmission rate. In the case of the same data requirements, the higher the transmission power, the shorter the entire flight time. As the data demand increases, the influence of the transmit power on the objective function

becomes more obvious. Therefore, we convert the objective function of minimizing time into maximizing throughput. The details are as follows.

$$(P10): \min_{P_u[n],\lambda_u[n],\xi_u[n]} \lambda_u[n]\delta_t$$

$$s.t. \sum_{n=1}^{N} \tilde{\chi}_u[n] \geq S_u, \forall u \tag{87}$$

$$\lambda_u[n] \leq \xi_u[n], \forall u, n \tag{88}$$

$$\left\| P_u[n], \xi_u[n], \left( \frac{2}{(\beta_0/\sigma_u^2)/((H^2 + \|L_u[n] - Z_u\|^2)^{\alpha/2})} \right)^{0.5} \right\| \leq P_u[n] + \xi_u[n], \forall u, n \tag{89}$$

$$0 \leqslant P_u[n] \leqslant P_{max} \tag{90}$$

$P_u[n], \lambda_u[n], \xi_u[n]$ can be updated through the problem (P10). Improve the ITE algorithm by changing the objective function in the problem (P8). The IMP Algorithm 2 is as follows.

---

**Algorithm 2** IMP Algorithm

---

**Initialize**: $L^0, P^0, I^0, \delta_t^0, r = 0$.
1: **Repeat**
2:       Solve problem (P3), and denote the optimal cache placement as $I_u^{f*}[n]$;
3:       Solve problem (P4), and denote the optimal trajectory as $L_u^*[n]$;
4:       Fix $L_u^*[n]$ and $I_u^{f*}[n]$, and $\{\lambda_u[n], \delta_t\}^r = \{\lambda_u[n], \delta_t\}^*$ by solving (P8),
          then denote the optimal transmission power as $\{P_u[n], \lambda_u[n], \delta_t\}^*$.
5: **Update** : the optimization variables and slack variables in $r$th iteration
6: **Update** : $r = r + 1$
7: **Until**:    Converge to a prescribed accuracy.
8: **Output**:  the Cache placement $I$, Transmission power $P$, UAV trajectory $L$ and
                completion time $T$.

---

## 6. Simulation and Discussion

### 6.1. Simulation Setup

This section verifies the feasibility and superiority of the algorithm through simulation experiments. In the simulation experiment, the UAV using the caching technology performs the data transmission task, and the users are randomly distributed in the given area. For both ITE and IMP algorithms, the initial trajectory of the UAV is flying in a straight line at a constant speed during the mission. In this section, the performance of the ITE algorithm and the IMP algorithm are compared by using the straight line flight optimization (SLF) [36] and the trajectory optimization scheme (TOS) [34]. In the expression of flight energy consumption of UAV, some parameters are assumed as follows: $U_{tip} = 120$, $v_0 = 4.03$, $A = 0.503$, $s = 0.05$, $\Omega = 300$, $k = 0.01$, $R = 0.4$, $\delta = 0.012$. Other parameters are shown in Table 1.

**Table 1.** Simulation parameters.

| Symbol | Explanation | Value |
|---|---|---|
| $d_0$ | The fuselage drag ratio | 0.6 |
| $\rho$ | The air density | 1.225 kg/m$^3$ |
| $P_{max}$ | UAV maximum speed | 900 MW |
| $\sigma_u^2$ | Noise power at the UAV | $-110$ dBm |
| $\beta_0$ | Channel power gain at the reference distance of 1 m | $-60$ dB |
| $H$ | The UAV altitude | 80 m |
| $V_{max}$ | Maximum UAV speed | 30 m/s |
| $F, C_r$ | Category and cache capacity | 50, 30 contents |
| $f$ | Size of each content | 1 Mbits |

*6.2. Simulation Results and Analysis*

In this paper, these users are randomly distributed in the area of 1 km $\times$ 1 km. Figure 2 shows the comparison of task completion time of four schemes for different data requirements. The ITE algorithm and the IMP algorithm are compared with the other two schemes. In Figure 2, when $S_u = 1$ Mb, the performance of the four algorithms is similar and the task completion time is roughly the same. This is because the data requirements are small and good performance can be achieved without any optimizations. At the same time, when $S_u = 1$ Mb, because there are many optimization variables in the ITE algorithm, compared with the other three schemes, its performance is poor and it takes a long time to complete the task. With the increase in the data requirement, the performance of the ITE algorithm and the IMP algorithm is better than the two external schemes. Compared with the other three algorithms, when the data demand is $S_u = 25$ Mb, the IMP algorithm has the best performance and the data transmission task time is the least.

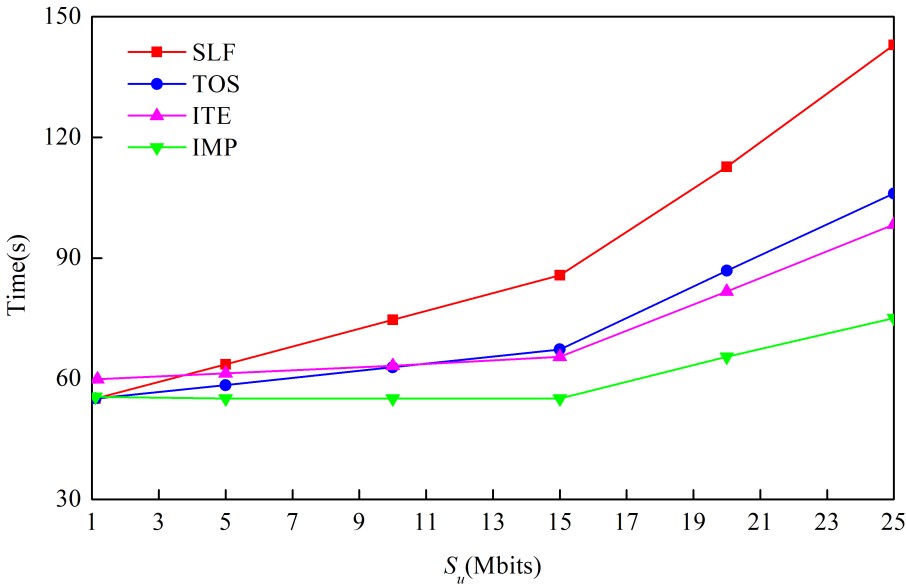

**Figure 2.** The relationship between task completion time and data requirement.

Next, we discuss the convergence of the ITE algorithm and the IMP algorithm in mission completion time and the UAV propulsion energy consumption. In Figure 3, the minimum data requirement of each user is set to $S_u = 15$ Mb. After multiple iterations, the task completion time and the energy consumption of the two schemes converge. From Figure 3a, it can be seen that when the UAV speed reaches the optimal value, the convergence speed of the ITE algorithm starts to slow down. However, the convergence speed of the IMP algorithm is still very fast. Similarly, for the recommended energy consumption, it can be seen from the Figure 3b that as the number of iterations increases,

the energy consumption of the IMP algorithm is lower than that of the ITE algorithm, and the convergence speed is faster than that of the ITE algorithm.

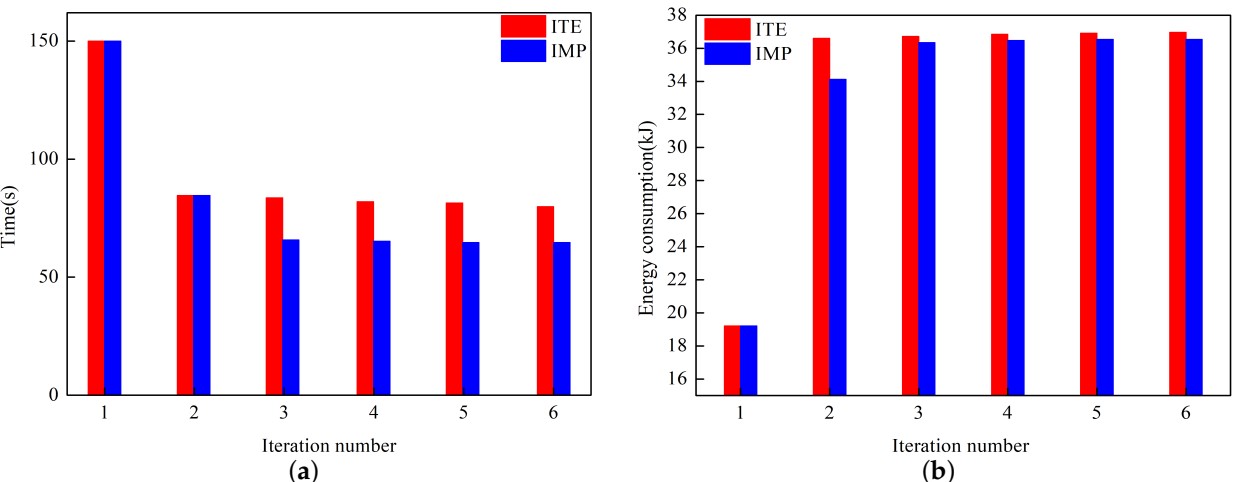

**Figure 3.** Convergence of ITE algorithm and IMP algorithm. (**a**) Convergence of the two algorithms in time. (**b**) Convergence of the two algorithms in energy consumption.

### 6.2.1. Comparison of Data requirements

According to different data requirements, the trajectory and speed of the UAV of the four schemes of the ITE algorithm, the IMP algorithm, the trajectory optimization scheme and the straight-line flight optimization scheme are compared. The results are shown in Figures 4 and 5. It can be seen from Figure 4 that when the data demand of each user is $S_u = 1$ Mb, the UAV chooses to fly straight during the entire data transmission process. When $S_u = 15$ Mb, the flight path of the UAV begins to approach the user. When $S_u = 25$ Mb, the UAV will pass the top of each user, because the closer the UAV is to the user, the higher the communication rate. When the data demand is low, for the other two algorithms, the TOS algorithm needs to be closer to the user. This is because, when the communication rate is high, the TOS algorithm only optimizes the trajectory of the UAV. The ITE algorithm and IMP algorithm proposed in this paper not only optimize the cache placement, but also optimize the UAV's transmission power and trajectory. Combining the Figures 4 and 5, it can be seen that when the data demand is low, the UAV will fly directly to the destination at a faster speed, minimizing the task completion time. When the data demand gradually increases, the UAV will gradually slow down to get closer to the user. When the data demand reaches $S_u = 25$ Mb, the UAV will have two states: flying at maximum speed or hovering on the user. When hovering above the user, the UAV will fly to the position with the best channel link at maximum speed, and then hover there for data transmission. It improves the communication efficiency of the network while reducing the completion time.

### 6.2.2. Comparison for Different Energy Constraints

Figures 6 and 7 show the impact of the three algorithms on the UAV's trajectory and speed when the value of $O_{max}$ is different. Set the data requirement of each user to $S_u = 15$ Mb. From the three figures in the Figure 6, it can be seen that the IMP algorithm is smoother than the ITE algorithm and the TOS algorithm. Under different energy constraints, compared with the other three algorithms, it can be seen that the higher the $O_{max}$, the more drastic the trajectory of the IMP algorithm changes. In Figure 7, the IMP algorithm is relatively stable during the entire flight mission completion process, and the mission completion time is the least. By comparing the three figures in the Figure 7, it can be found that the higher the $O_{max}$, the higher the speed of the UAV, which is why the trajectory of the UAV becomes sharper.

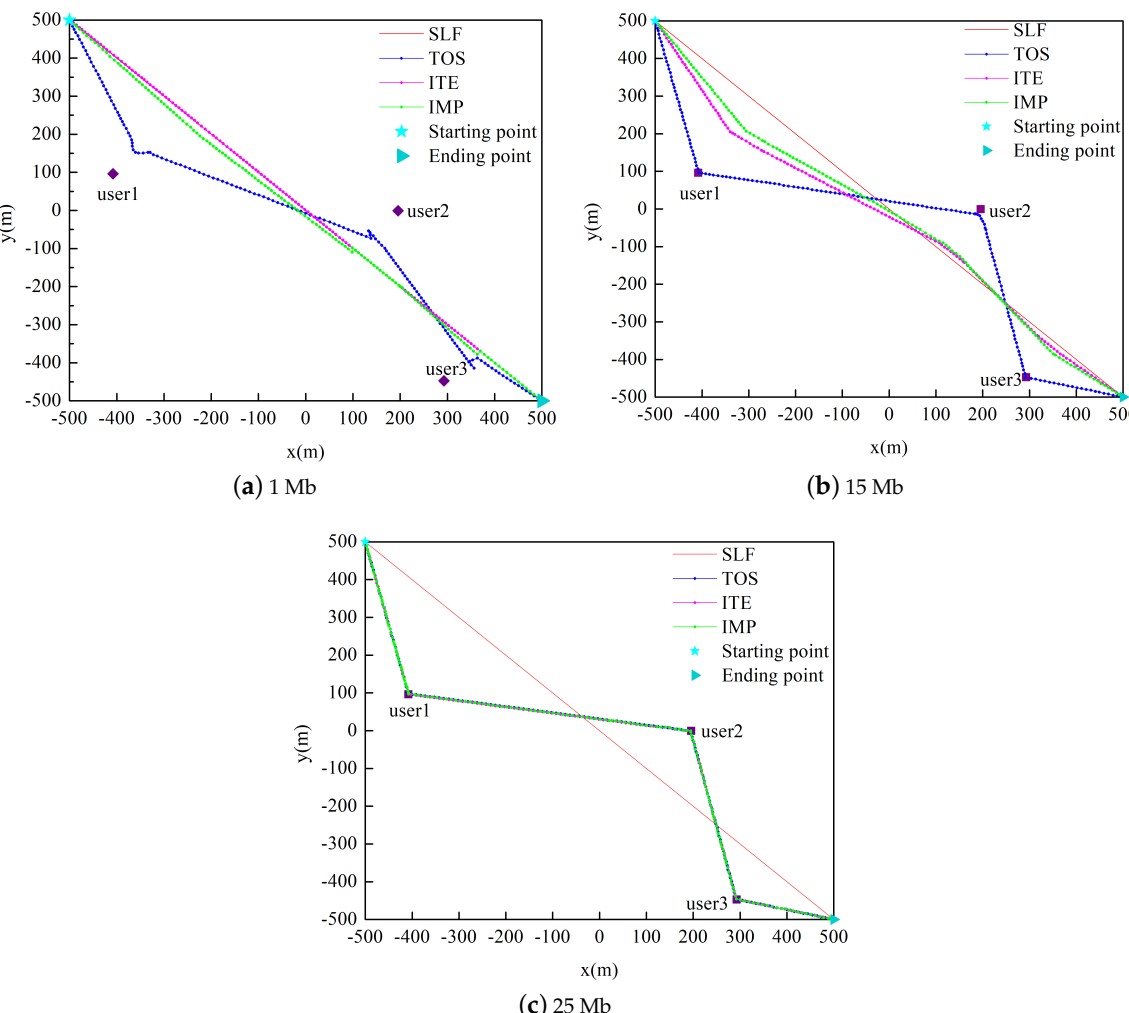

**Figure 4.** According to different data requirements, the trajectory comparison of UAV.

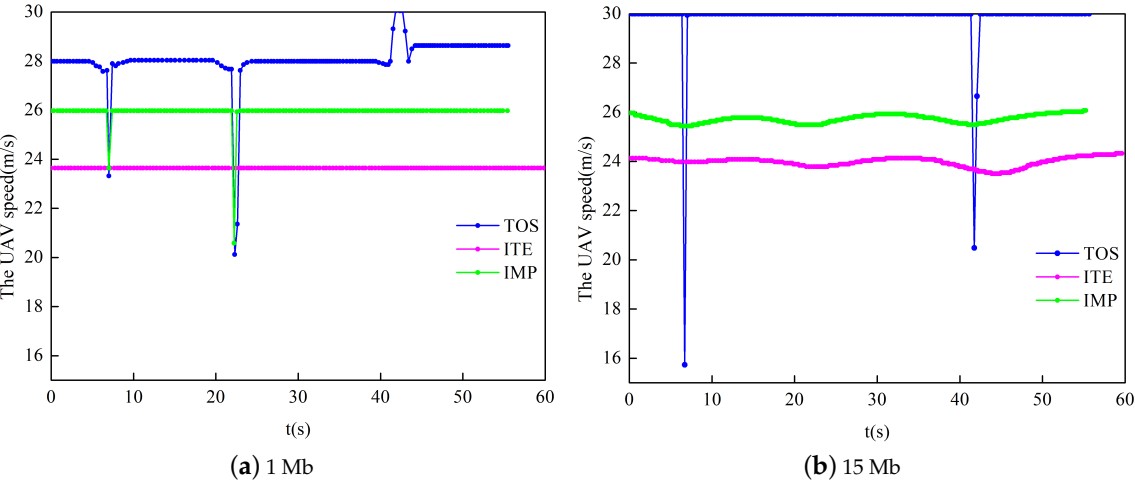

**Figure 5.** *Cont.*

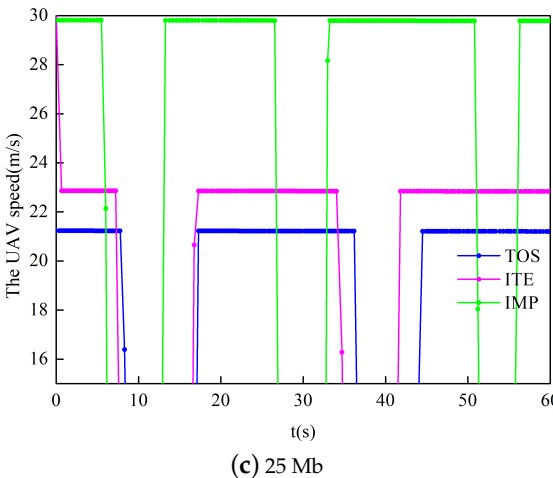

(**c**) 25 Mb

**Figure 5.** According to different data requirements, the speed comparison of UAV.

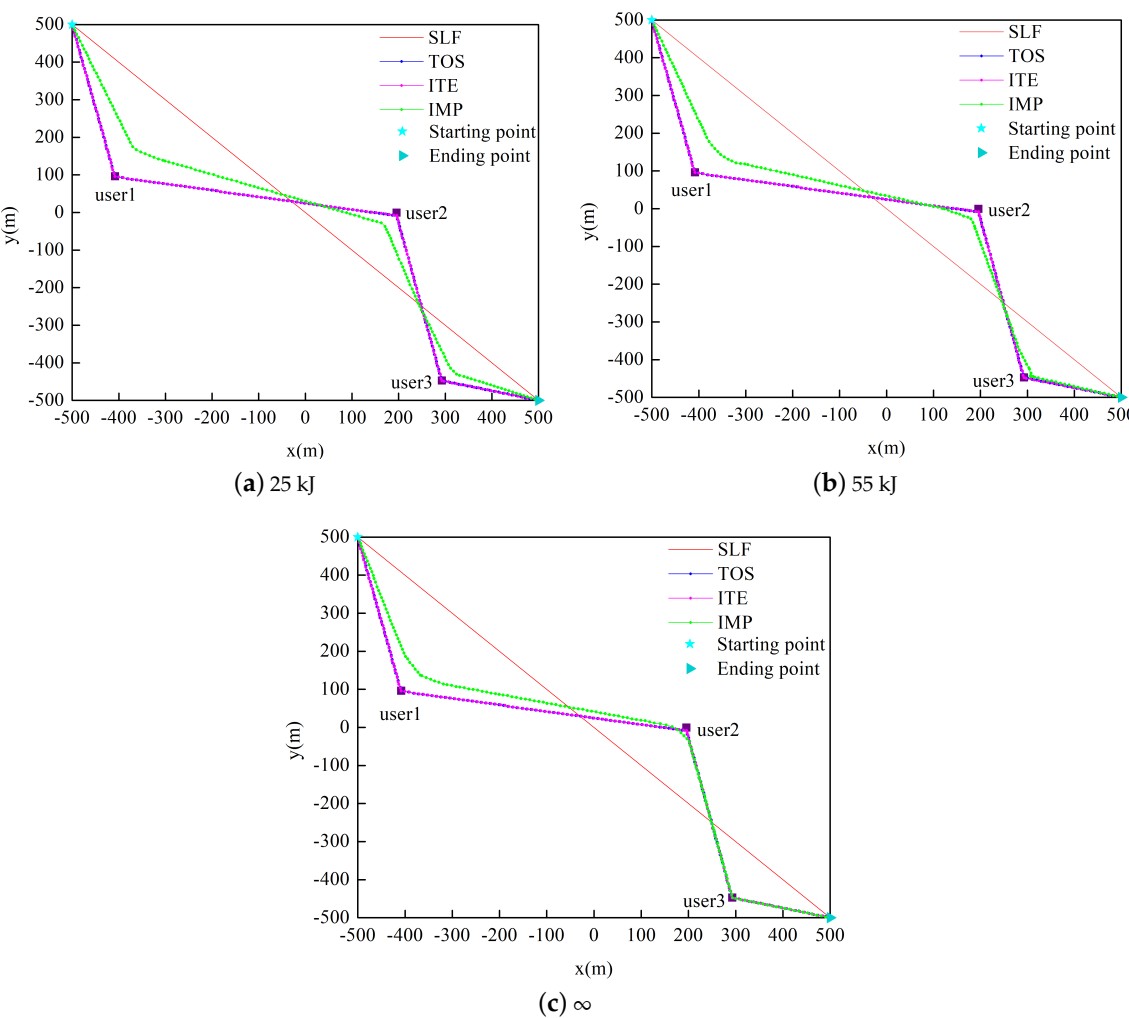

(**a**) 25 kJ

(**b**) 55 kJ

(**c**) ∞

**Figure 6.** According to different energy constraints, the trajectory comparison of UAV.

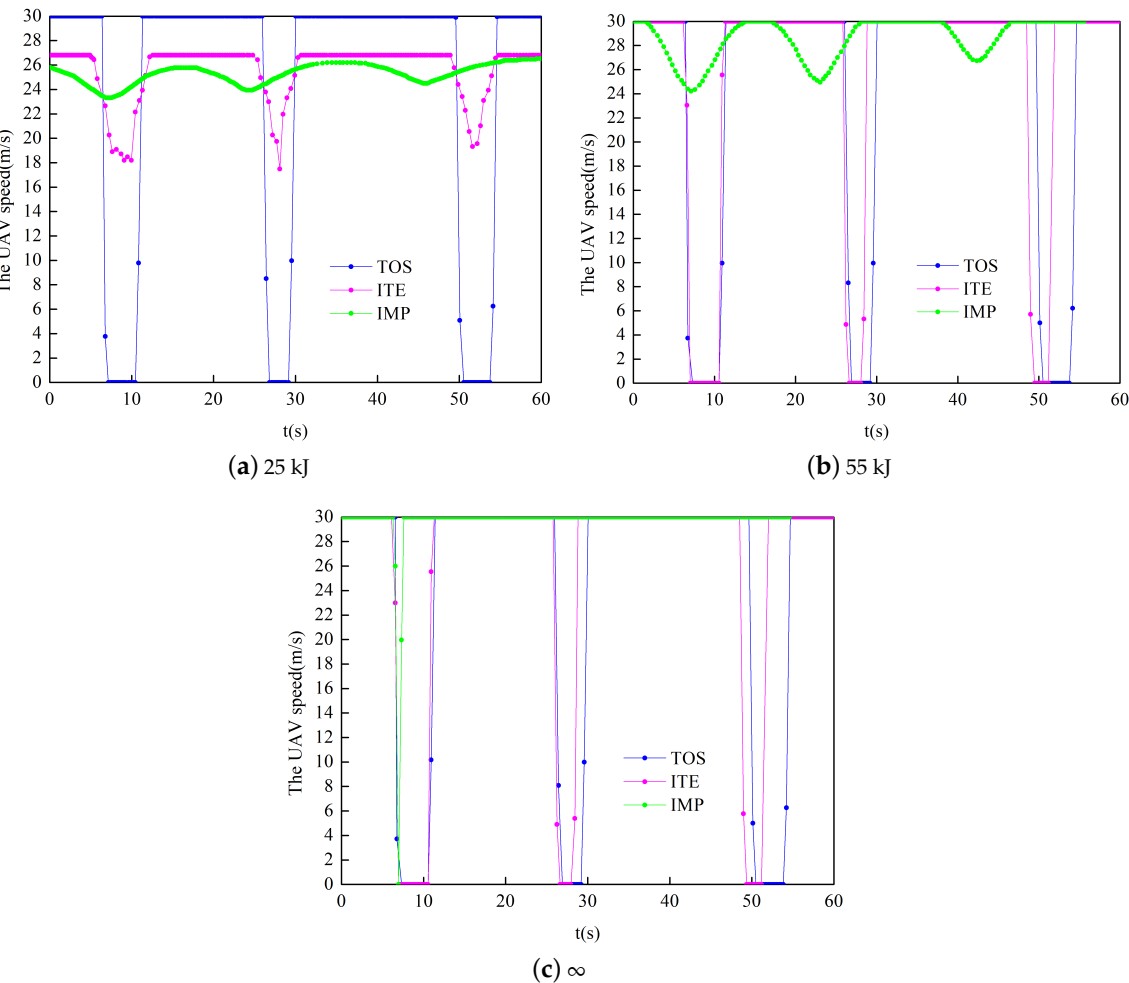

**Figure 7.** According to different energy constraints, the speed comparison of UAV.

Figure 8 shows the trajectory and speed of the UAV under different energy constraints of the IMP algorithm. From the Figure 8a, it can be seen that when $O_{max} = 25$ kJ, the trajectory of the UAV is smoother than the trajectory under other energy constraints. It can be seen from the Figure 8b that the higher the energy constraint $O_{max}$, the higher the speed of the UAV and the shorter the completion time. Conversely, the slower the UAV, the longer it will take to complete.

### 6.2.3. Transmit Power Allocation for the IMP Algorithm

In this paper, these users are randomly distributed in the given area. When $S_u = 25$ Mb, the power allocation of the UAV in the IMP algorithm is shown in Figure 9. During the entire flight, the UAV will gradually allocate power to users close to it until the user's power reaches the maximum. As the UAV begins to slowly move away from the user, the power allocated to the user will gradually decrease until the power drops to 0. At the same time, the UAV will gradually allocate power to users close to it until the user's power reaches the maximum.

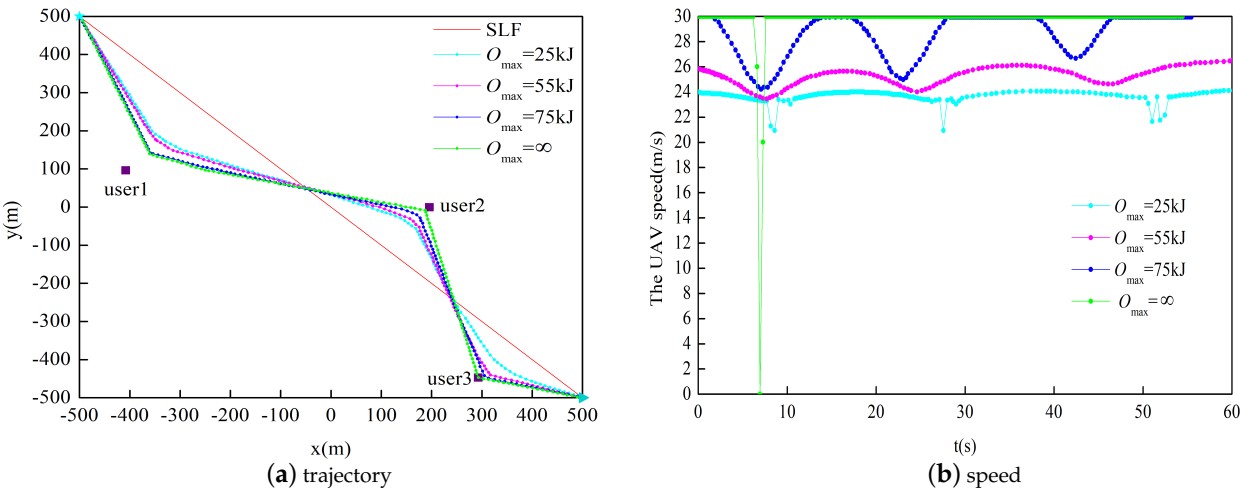

**Figure 8.** Under different energy constraints, the trajectory and speed of the UAV after adopting the IMP algorithm.

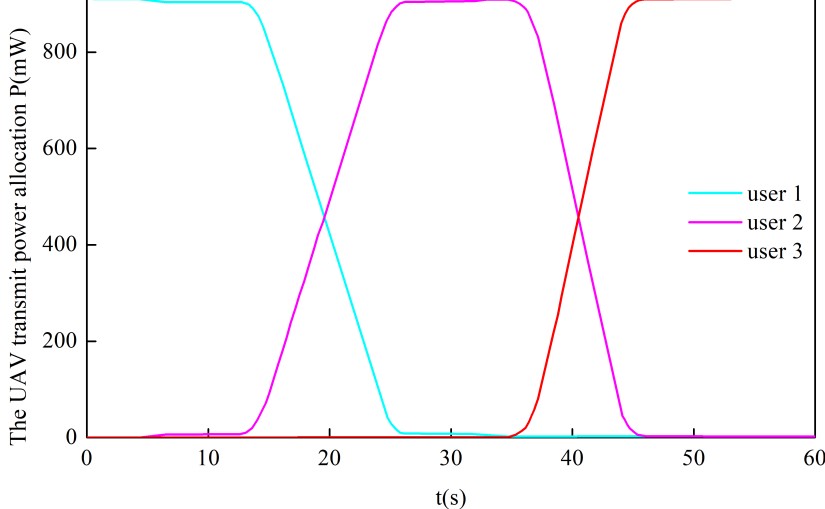

**Figure 9.** The relationship between UAV power allocation and time.

6.2.4. Trajectories for Large-Scale Scenarios

Next, we will verify the performance of the IMP algorithm when the UAV sends data to large-scale users, where the user equipment is randomly distributed in the area of 2 km × 2 km. Figure 10 shows the UAV's trajectory when the data demand of each user is $S_u = 1$ Mb. It can be seen from the figure that the UAV tends to slowly approach the user equipment far away from itself, which can reduce the path loss caused by the long-distance, but will not pass through each user equipment. There is a trade-off between communication rate and completion time to meet minimum data requirements and minimize completion time. The difference from Figure 10a,b is that the UAV returns to the origin after the mission is completed. It can be observed that the UAV is flying in an area with dense user equipment, and the first half and the second half of the UAV's flight trajectory are symmetrical. In order to shorten the completion time as much as possible and meet the minimum data requirements of all user equipment, the UAV tends to allocate half of the data in the first flight and half of the data in the second flight.

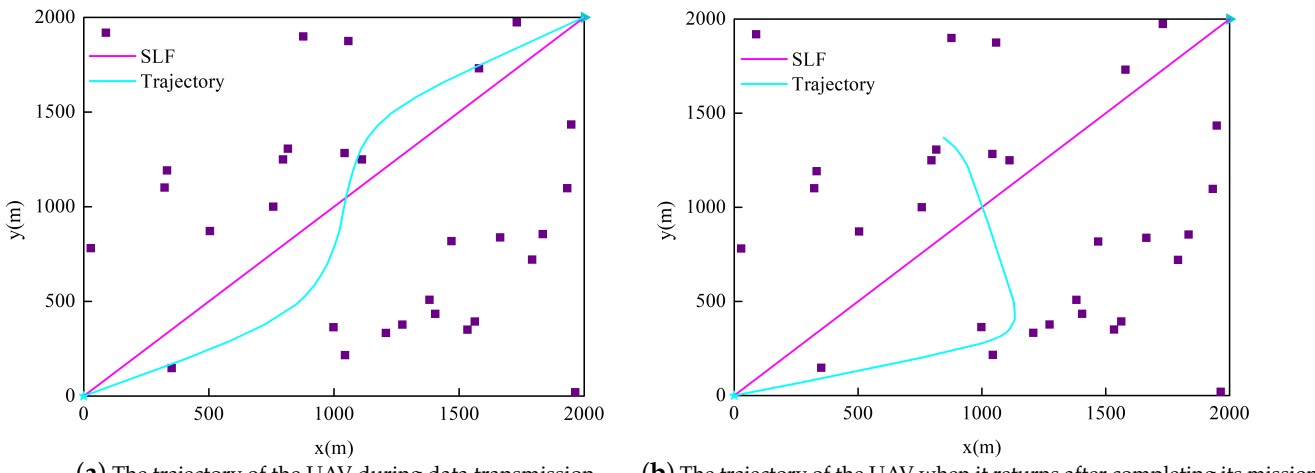

(**a**) The trajectory of the UAV during data transmission

(**b**) The trajectory of the UAV when it returns after completing its mission

**Figure 10.** UAV trajectory in a large-scale scenario.

### 6.2.5. UAV Mission Completion Time for Different $\varsigma$ Values

Figure 11 shows the effect of different $\varsigma$ values on the completion time of the UAV mission. It can be seen from the figure that when the UAV cache capacity is the same, the larger the $\varsigma$, the shorter the task completion time. This is because $\varsigma$ represents the skewness of content popularity. The larger the distribution of popular files, the more concentrated the UAV cache hit rate, and the higher the UAV cache hit rate. The content requested by the user is easier to obtain from the UAV without passing through the transmission link from the BS to the UAV. Therefore, the higher transmission performance of the system will reduce the mission completion time of the UAV.

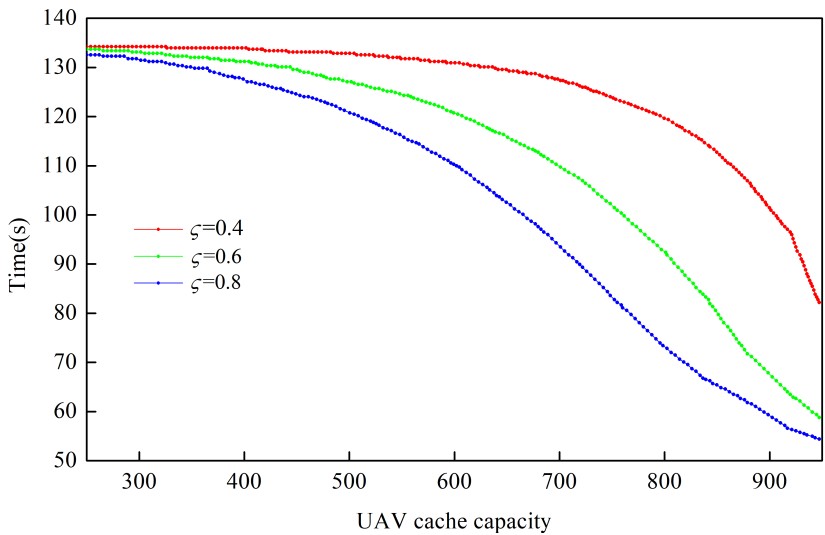

**Figure 11.** Effect of different $\varsigma$ values on UAV mission completion time.

### 7. Conclusions

In this paper, a UAV-assisted wireless communication system using caching technology is studied. The UAV can pre-store a part of popular content to provide users with data transmission services. Considering the limited storage space, the goal is to minimize the time for the UAV to serve users by jointly optimizing cache placement, the UAV's trajectory and transmission power under the constraints of maximum energy estimation and data requirements for each user. In order to solve this non-convex optimization problem, an iterative algorithm based on successive convex approximation and alternating optimization techniques is proposed. In addition, we have also improved the proposed iterative algo-

rithm to improve the performance of the algorithm. The simulation results show that the performance of the proposed algorithm is verified by comparing with various benchmark schemes. We can also carry out many other research directions in our future work. For example, in the field of UAV-assisted edge computing. Considering several ground users with limited computing power in the target area, when the computing tasks they face exceed their computing power, they use the UAV to deploy edge computing servers over the target area to unload data. At this time, it is necessary to consider the communication rate between the user and the UAV, the UAV computing capability, the calculation data offloading strategy, and the UAV's trajectory optimization problem.

**Author Contributions:** Tingting Lan conceived the presented idea. Tingting Lan and Danyang Qin developed the theory and performed the computations. All authors discussed the results and contributed to the final manuscript. All authors have read and agreed to the published version of the manuscript.

**Funding:** This research was funded by Support by the National Natural Science Foundation of China (61771186), Outstanding Youth Project of Provincial Natural Science Foundation of China in 2020 (YQ2020F012), Nursing Program for Young Scholars with Creative Talents in Heilongjiang Province (UNPYSCT-2017125), Distinguished Young Scholars Fund of Heilongjiang University, postdoctoral Research Foundation of Heilongjiang Province (LBH-Q15121), and Postgraduate Innovative Research Project of Heilongjiang University (YJSCX2021-172HLJU).

**Data Availability Statement:** The data presented in this study are available on request from the corresponding author. The data are not publicly available due to privacy.

**Acknowledgments:** We thank the reviewers for the thorough review and greatly appreciate the comments and suggestions, which significantly contributed to improving the quality of the publication.

**Conflicts of Interest:** The authors declare no conflicts of interest.

## Abbreviations

The following abbreviations are used in this manuscript:

| | |
|---|---|
| UAV | Unmanned Aerial Vehicles |
| BS | Base Station |
| SCA | Successive Convex Approximation |
| LoS | Line of Sight |
| SNR | Signal Noise Ratio |
| FoT | First-order Taylor |

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
