# Peer review of "Joint Optimization on Trajectory, Cache Placement, and Transmission Power for Minimum Mission Time in UAV-Aided Wireless Networks"

_ijgi, doi:10.3390/ijgi10070426_

Round 1

Reviewer 1 Report

This paper presents solution to three problems with UAV: trajectory optimization, cache placement optimization and power allocation optimization.

I have several comments:

  • Please start section's numbering from 1.
  • Introduction should be finished with one paragraph that describe what are the sections of the paper
  • There is a lot of mathematics so it is quite hard to follow. I wonder if you implemented these algoritms in some programming language?
  • Also did you try those algorithms in practice?
  •  

Reviewer 2 Report

The authors have done a lot of work with the noble goal of developing a methodology to optimize the use of UAV resources.
Overall, impressions of the work are positive, but in the context of the goals and mission of the ISPRS International Journal of Geo-Information, it is not entirely clear what conclusion can be drawn from the results of the work regarding the acquisition of GIS-ready information.
Again, when using a UAV to survey an area for geoinformation purposes, the high speed of transmission of information wirelessly immediately to the base station is not so important - the UAV will not fly over such an area at one time anyway so that the footage will not fit into the volumes of the built-in data storage. Control and speed of signal transmission from the remote control to the rover are also not important, because the UAV flies according to a predetermined (taking into account altitude and longitudinal and transverse overlap of neighboring tacks) flight task. Another thing is an optimization of the trajectory - but in this case, the trajectory is important only when the UAV starts to the beginning of the flight task: the starting climb is the most energy-consuming stage of the flight, in the case of a "copter-type" UAV the climb can take up to 5% of the battery volume. An altitude gain with simultaneous horizontal movements would help save a few percent of the charge.
Therefore, it is necessary to modify the manuscript so that the purpose of the work is suitable for solutions in the field of geographic information systems.

Reviewer 3 Report

The authors present a complex method to optimize three important aspects in UAV-assisted wireless networks (trajectory, cache placement, and transmission power) with the aim of minimizing mission time. The structure of the paper is adequate. Related work needs to be improved. The tests carried out could be explained in more detail. English is correct. Some comments for improvement of the paper are below.

C1. Abstract. It is not necessary to include the acronyms BS and ITE because they are not cited in the abstract.
C2. Introduction. At the end of the introduction it is necessary to include the contributions of the authors and the structure of the paper.
C3. Related work. This part of the paper looks like a compilation of references without any classification. Some classification from other authors' proposals for trajectory, cache placement, and minimum transmit power should be included. On the other hand, some of the papers are very recent, which has implied a great deal of work on the part of the authors.
C4. Line 113 and following. These paragraphs are the solution to comment 2. It should not be in this section.
C5. System model. The time is missing in the expression V (t) = L (t) as indicated in equation 1. Right after 'in adition'.
C6. Channel model. Include a reference that uses the same model.
C7. Cache placement model. Include a reference to the Zipf distribution that answers the question: why is this distribution used?

C8. Energy consumption model. Include some reference to the model used.
C9. Energy consumption model. Is the energy consumption of communications negligible? Include a reference that makes a study of this concept.
C10. Problem formulation. Is all the data that you need to have prior to the simulation? The iterative algorithm must be offline, but most situations cannot be known in advance to carry out this process. Is there a way to use this algorithm online?
C11. Page 10. Why is FOT used? Why not the second order of Taylor?
C12. Simulation setup. References for SLF and TOS are required.
C13. Section 5.2 It is necessary to have an introduction prior to the different tests in order not to get lost in the text.
C14. Section 5.2 Only three users? Such a complex procedure to test with only three users does not seem profitable.
C15. Table 1. How do you know the demand of each user? How do you know the data that each user will ask for? Where are the users located? How long is the simulation? Include more data to try to make the experiment reproducible. 
